# Recent Progress of Oral Functional Nanomaterials for Intestinal Microbiota Regulation

**DOI:** 10.3390/pharmaceutics16070921

**Published:** 2024-07-10

**Authors:** Wanneng Li, Minle Zhan, Yue Wen, Yu Chen, Zhongchao Zhang, Shuhui Wang, Dean Tian, Sidan Tian

**Affiliations:** 1Department of Gastroenterology, Tongji Hospital, Tongji Medical College, Huazhong University of Science and Technology, Wuhan 430074, China; m202276234@hust.edu.cn (W.L.); wenyue@tjh.tjmu.edu.cn (Y.W.); d202181940@hust.edu.cn (Y.C.); tjhzzc@hust.edu.cn (Z.Z.); d202282081@hust.edu.cn (S.W.); 2National Engineering Research Center for Nanomedicine, College of Life Science and Technology, Huazhong University of Science and Technology, Wuhan 430074, China; m202172338@hust.edu.cn; 3Key Laboratory of Molecular Biophysics of the Ministry of Education, College of Life Science and Technology, Huazhong University of Science and Technology, Wuhan 430074, China

**Keywords:** nanomaterial, gut microbiota, therapy, immunity, oral administration

## Abstract

The gut microbiota is closely associated with human health, and alterations in gut microbiota can influence various physiological and pathological activities in the human body. Therefore, microbiota regulation has become an important strategy in current disease treatment, albeit facing numerous challenges. Nanomaterials, owing to their excellent protective properties, drug release capabilities, targeting abilities, and good biocompatibility, have been widely developed and utilized in pharmaceuticals and dietary fields. In recent years, significant progress has been made in research on utilizing nanomaterials to assist in regulating gut microbiota for disease intervention. This review explores the latest advancements in the application of nanomaterials for microbiota regulation and offers insights into the future development of nanomaterials in modulating gut microbiota.

## 1. Introduction

The gut microbiota is a microbial community residing in the human intestinal tract, comprising thousands of different bacterial species, with the number of microorganisms in the human gut ranging from 10^14^ to 10^15^ [1,2]. In the gut, the normal composition of the gut microbiota is crucial for maintaining intestinal functions such as peristalsis and facilitating normal nutrient metabolism [3,4]. Additionally, metabolites produced by the gut microbiota can interact with intestinal mucosal cells to regulate host immune responses [5]. Outside the gut, the vast microbial composition also influences various aspects of human health, including growth and development, aging, regulation of the skeletal system, and balance of the immune system (Figure 1) [6,7,8,9]. Some products secreted by the gut microbiota can enter the bloodstream through the intestinal barrier, thereby affecting extraintestinal organs [10]. Therefore, strategies to manipulate the gut microbiota to treat certain diseases have garnered increasing attention from researchers. Currently, the most common interventions for modulating the gut microbiota include the following two major strategies: (1) oral administration of microbiota-modulating drugs and (2) fecal microbiota transplantation. Fecal microbiota transplantation involves transplanting functional microbiota from healthy donors into the patient’s gut to restore the composition of the gut microbiota and thus treat diseases. Fecal microbiota transplantation emphasizes the overall role of the gut microbiota rather than specific components. Studies have shown that the cure rate of fecal microbiota transplantation for *Clostridium difficile* infection can reach 90%. However, this treatment method presents many challenges compared to oral probiotics or medication for modulating the gut microbiota, such as gastrointestinal discomfort, mechanical damage, and low patient acceptance [11,12]. Among these, oral administration is undoubtedly the most common and convenient method of delivery, which can be entirely reliant on the patient and thereby enhance patient acceptance and compliance [13]. Oral administration not only has minimal side effects but also can reduce the systemic toxicity of colon-targeted therapy and improve its efficacy [14]. However, oral administration also faces many challenges. Most of the gut microbiota colonize the colon, so when using oral probiotics to intervene in the disease process, probiotics need to pass through the oral cavity, esophagus, stomach, small intestine, and colon [1]. During this process, probiotics undergo prolonged transportation and endure changes in the surrounding environmental pH and damage from digestive fluids, which cannot guarantee the colonization and activity of probiotics, leading to an increase in the effective dose of probiotics [15]. Additionally, many researchers have indicated that oral probiotics are not entirely safe, but currently, there is a lack of research on the safety of oral probiotics [15,16,17]. Oral antibiotics are currently commonly used to eliminate harmful gut microbiota. However, due to their uncontrollable effects on other normal microbiota, this method is not conducive to maintaining the balance of the gut microbiota [6,18]. Therefore, improving the application of oral administration systems in the regulation of the gut microbiota is currently an urgent issue. The combination of nanomaterials and oral administration systems is a promising strategy. On the one hand, nanoparticles as carriers can transport microbiota-modulating drugs to prevent them from physical and chemical damage from the intestines. Moreover, specific nanomaterials can also help in the controlled release and targeted delivery of drugs by responding to changes in intestinal chemistry. On the other hand, nanomaterials themselves can also participate in microbiota regulation.

Nanomaterials refer to materials where at least one dimension is in the nanoscale range (1–100 nm). The rapid development of nanotechnology has greatly promoted the application of nanomaterials in biomedicine, especially in oral formulations. Compared to other administration methods, oral administration is undoubtedly the most convenient way to administer drugs. However, traditional drugs often have low bioavailability due to the influence of their properties and gastrointestinal environments. However, traditional drugs often have low bioavailability due to the influence of their properties and gastrointestinal environments [19]. Oral nanocarriers carrying microbiota-modulating drugs can protect the drugs from damage by low pH in the stomach and intestinal enzymes, thereby increasing their bioavailability [20]. Furthermore, some nanocarriers can assist in the targeted transport of microbiota-modulating drugs, thereby increasing the concentration and activity of orally administered drugs at the lesion site, reducing the dosage of drugs used, and improving the safety of drugs. For example, materials with pH responsiveness can achieve the colon-targeted release of drugs [21]. Additionally, the combination of nanomaterials and antibiotics can not only maintain the balance of the gut microbiota but also reduce the emergence of drug-resistant bacteria [22]. As mentioned earlier, the influence of the gut microbiota on human health has received increasing attention, and intervention in the gut microbiota through regulation has become a new treatment strategy. However, the regulation of the gut microbiota is constrained by gastrointestinal environments and physiological functions. Fortunately, the application of nanotechnology has greatly increased the feasibility of regulating the gut microbiota. This paper summarizes the regulation of the gut microbiota by nanotechnology as carriers and nanomaterials themselves, the adverse effects of certain nanomaterials on the gut microbiota, and the application of nanotechnology in the treatment of diseases by regulating the gut microbiota, and finally puts forward further prospects for the regulation of the gut microbiota by nanomaterials.

## 2. Nanomaterials as Drug Carriers to Assist Microbiota Regulation

### 2.1. Nanomaterials as Carriers for Sustained Release of Microbiota-Modulating Drugs

Nanomaterials can serve as excellent carriers to transport certain microbiota-modulating molecules into the intestine to better exert their regulatory effects (Figure 2A) (Table 1). Polyphenols, compounds found in plant-based foods, have potential health-promoting effects and can interact with targets in the body, exhibiting anti-inflammatory and antioxidant effects [23]. Among them, low-toxicity and health-promoting polyphenolic substances are gradually being explored by researchers as good food additives [24]. These substances can interact with the gut microbiota, and some polyphenols can alter the composition of the gut microbiota, promoting the growth of beneficial bacteria while inhibiting the proliferation of harmful bacteria [25]. Although polyphenolic substances have various excellent biological activities, their low solubility, poor digestibility, and rapid absorption and metabolism during gastrointestinal digestion processes result in their less-than-ideal bioavailability. The sustained-release function of nanomaterials can effectively compensate for the low bioavailability of polyphenols. Wang Luet al. constructed nanoparticles composed of Hohenbuehelia serotina polysaccharides and mucins as carriers to encapsulate polyphenols from *Malus baccata (Linn.) Borkh* (MBP-MC-HSP NPs). This nanocarrier exhibited good sustained-release effects, effectively protecting polyphenols, isolating them from the biological effects of the intestinal tract, and presenting no toxic side effects. Experimental results showed that MBP-MC-HSP NPs indeed promoted intestinal health through microbiota regulation [24]. Similarly, Mejo Kuzhithariel Remanan et al. encapsulated chrysin and rutin (natural polyphenols) using starch nanoparticles formed by the self-assembly of starch from quinoa (Q), maize (M), and waxy maize (WM). Among them, starch nanoparticles formed by quinoa showed sustained release of polyphenols and enhanced the antioxidant activity of polyphenols [26]. Xiaoyu Li et al. reported the design and manufacture of nanoparticles based on Hohenbuehelia serotina polysaccharides (HSP) and bovine serum albumin (BSA) as carriers for delivering polyphenols isolated from the shells of Juglans regia L (BSA-JRP-HSP NPs). BSA-JRP-HSP NPs promoted the growth of probiotics, inhibited the proliferation of harmful bacteria, and significantly increased the content of short-chain fatty acids during colonic fermentation [27]. Additionally, Wenni Tian et al. used nanostructured lipid carriers (6G-NLCs) and NLC-imbedded microcapsules (6G-MCs) loaded with 6-gingerol, respectively. Different delivery materials have distinct biological distributions and release curves, even when loaded with the same 6-gingerol, which can lead to different intervention results on the gut microbiota [28]. The pattern diagrams demonstrate the release of 6G from 6G-NLCs and 6G-MCs in the stimulated gastric fluid (SGF), stimulated intestinal fluid (SIF), and stimulated colon fluid (SCF) in vitro. Among them, the 6G-NLC showed a very small amount of release in the SIF and was completely released in the SCF. However, the 6G-MC only showed release in the SCF (Figure 2B). Meanwhile, Wenni Tian et al. observed the structural evolution of the 6G-MC during in vitro digestion using light microscopy. In the SGF and SIF, the structure of the 6G-MC remained intact. After placing 6G-MCs in the SCF, the figure shows sustained destruction of its structure over a period of 5 h (Figure 2C). According to the authors in the original article, 94.32% of the 6G in the SCF would be released continuously within 20 h [28]. In summary, the use of nanomaterials for delivering polyphenols can significantly improve their bioavailability and assist in their intervention on the gut microbiota. Nanomaterials are not merely delivery tools for polyphenols; the differences in delivery materials can significantly affect the intervention mechanism and efficacy of polyphenolic substances on the gut microbiota. The sustained-release effect of nanomaterials can also be applied to the combination with small-molecule water-soluble drugs. Pengchao Zhao et al. reported a biologically adhesive liquid coacervate assembly based on hydrogen-bond-driven nanoparticles, which enhanced the retention time of Sodium phosphate salt of dexamethasone (small-molecule water-soluble substance) in the intestine, prolonged its action time, reduced the dosage of this drug, and thus reduced its side effects. Additionally, since the balance of intestinal inflammation is complementary to the stability of the gut microbiota, this nanodrug also has a reparative effect on the gut microbiota while exerting anti-inflammatory effects [29].

### 2.2. Nanomaterials as Carriers for Controlled Release of Microbiota-Modulating Drugs

The intestinal system is a complex structure, and the effects of oral drugs on the gut microbiota are often influenced by the acidity and enzymatic environment of the digestive tract. The application of nanocarriers can overcome this challenge and, at the same time, control the release of drugs to ensure their action at the damaged sites of diseases as much as possible (Figure 3A) (Table 1) [30,31]. Shangyong Li et al. developed an enzyme-triggered controlled-release system with curcumin-cyclodextrin (CD-Cur) complexes as the core and low-molecular-weight chitosan and unsaturated alginate nanoparticles (CANP) as shells. The pH-sensitive and α-amylase-responsive release characteristics of this nanomaterial endowed it with excellent colon-targeting ability and controlled-release capability. This greatly enhanced the curative effect of curcumin on colitis and promoted the restoration of gut microbiota composition [32]. As shown, the release of curcumin (Cur) from Cur-CD-CANPs varied in response to changes in PH and enzyme environments. In a low PH, α-amylase-free environment such as the SGF, the release of Cur is relatively low, which protects Cur from destruction. On the contrary, at relatively high pH and in the presence of α-amylase, such as the SIF, the release of Cur is significantly increased, which can make Cur better utilized by the body (Figure 3B) [32]. Probiotics are a class of beneficial active microorganisms that colonize the human body and affect human health in a favorable way [33]. As mentioned earlier, although oral probiotics have been confirmed as one of the means to promote health and improve diseases, due to the complex physiological and chemical structure of the intestine, their stable arrival at disease sites cannot be guaranteed. Jiali Chen et al. reported a supramolecular self-assembled nitroreductase (NTR) labile peptidic hydrogel and used it to encapsulate the typical probiotic *Escherichia coli Nissle 1917* (EcN). This nanocomposite not only protected probiotics but also, because of the upregulated nitroreductase in the intestine, which can trigger the decomposition of the hydrogel, enabled the probiotics to be released locally under controlled conditions, significantly improving the treatment of inflammatory bowel disease [34].

### 2.3. Nanomaterials Assisting in Targeted Microbiota-Modulating Drug Delivery

Nanomaterials can assist drugs in targeting disease sites, enriching drugs at lesion sites, and increasing drug efficacy while reducing side effects (Table 1). Macrophage inflammatory chemotaxis capability can endow nanocomposites with a targeted ability to inflammation sites, increasing drug concentrations at inflammatory sites. Based on this, Meiyu Bao et al. utilized macrophage membranes to encapsulate coupled complexes of polydopamine nanoparticles with mouse cathelicidin-related antimicrobial peptide(mCRAMP), thereby conferring their inflammation-targeting ability. Oral administration of the nanocomplexes significantly reduced colonic inflammation and positively regulated the intestinal microbiota in a mouse model of colitis (Figure 4A) [35]. The PEGylated poly (α lipoic acid) (PEG-PALA) copolymer nanoparticles can achieve antibiotic-targeted delivery in response to hydrogen sulfide released by *Salmonella*, promoting targeted therapy for *Salmonella* infection (Figure 4B). This nanomaterial not only reduces Salmonella colonization in the body but also significantly maintains gut microbial homeostasis [36]. As shown in the figure, in the SGF and SIF, the release of ciprofloxacin (CIP) is low, whereas the addition of NaS significantly promotes the release of CIP and the amount and rate of CIP release increase with increasing concentrations of added NaS (Figure 4D). Meanwhile, the authors treated *Salmonella Typhimurium ATCC 14028, Escherichia Coli ATCC 25922, Klebsiella Pneumoniae ATCC 700603,* and *Enterobacter Cloacae ATCC 13047* in vitro with free CIP and PEG-PALA@CIP, respectively. It was found that PEG-PALA@CIP showed a stronger inhibitory effect on *S. typhimurium ATCC 14028*, which can produce H_2_S, than the other bacteria (Figure 4E) [36]. Iron-based ions such as Fe@Fe_3_O_4_ nanoparticles’ self-liver targeting function can assist nanodrugs in targeting liver cancer treatment. Zhigang Ren et al. developed a liver cancer-targeted therapeutic nanomedicine using the coupling of Fe@Fe_3_O_4_ and ginsenoside Rg3, which can inhibit the development and metastasis of hepatocellular carcinoma by remodeling the imbalanced intestinal microbiota [37]. pH-responsive materials such as rhamnolipid can endow nanomaterials with a colon-targeting ability. Based on this property, Yuxuan Xia et al. developed rhamnolipid (RL)/fullerene (C_60_) nanocomposites with a colon-targeting ability, which significantly alleviated the inflammation in a mouse model of colitis while restoring the intestinal microbes to near-normal levels in mice [21]. Xiaochen Su et al. developed a highly cross-linked polyphosphazene nanodrug that not only enhanced the bioavailability of polyphenolic substances like curcumin but also achieved targeted release of curcumin for inflammation, alleviating acute lung injury [38]. Oral silicon hydrogen nanoparticles can be adsorbed onto the inflamed epithelium of the colon by electrostatic action to target the site of colonic inflammation and alleviate inflammatory bowel disease by scavenging reactive oxygen species (ROS) at the site of the lesion as well as positively regulating the intestinal microbiota (Figure 4C) [39].

**Figure 3 pharmaceutics-16-00921-f003:**
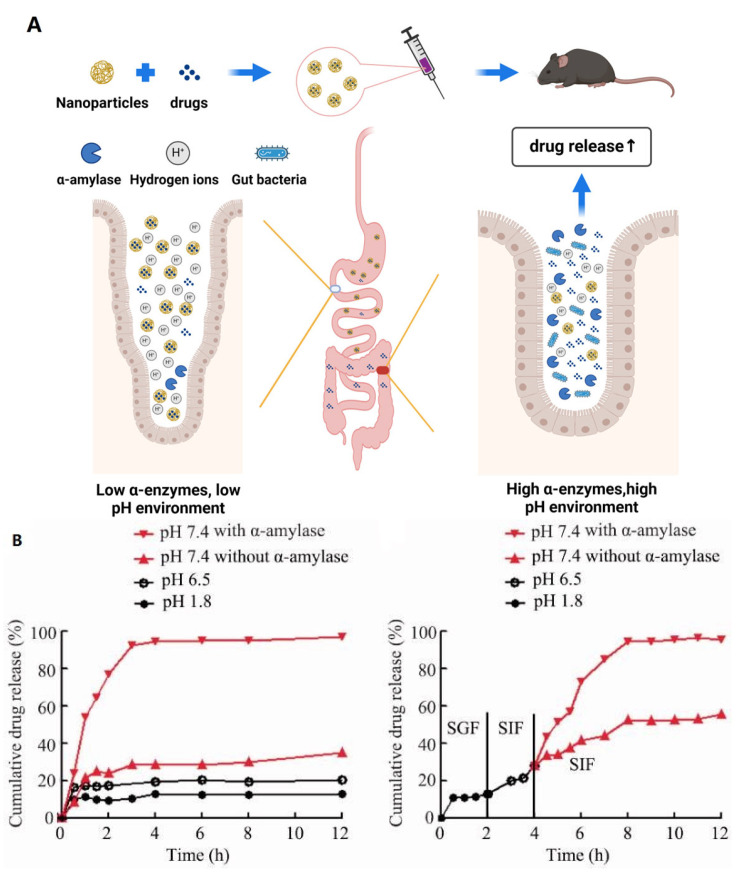
(**A**) Schematic representation of nanomaterial-controlled drug release. (created with bioRender.com) (**B**) pH-dependent and enzyme-responsive drug release curves of CD-Cur-CAMP (an enzyme-triggered controlled-release system with curcumin-cyclodextrin (CD-Cur) complexes as the core and low-molecular-weight chitosan and unsaturated alginate nanoparticles (CANP) as shells) in vitro. Adapted with permission from ref. [32]. Copyright 2021 Informa Healthcare.

**Figure 4 pharmaceutics-16-00921-f004:**
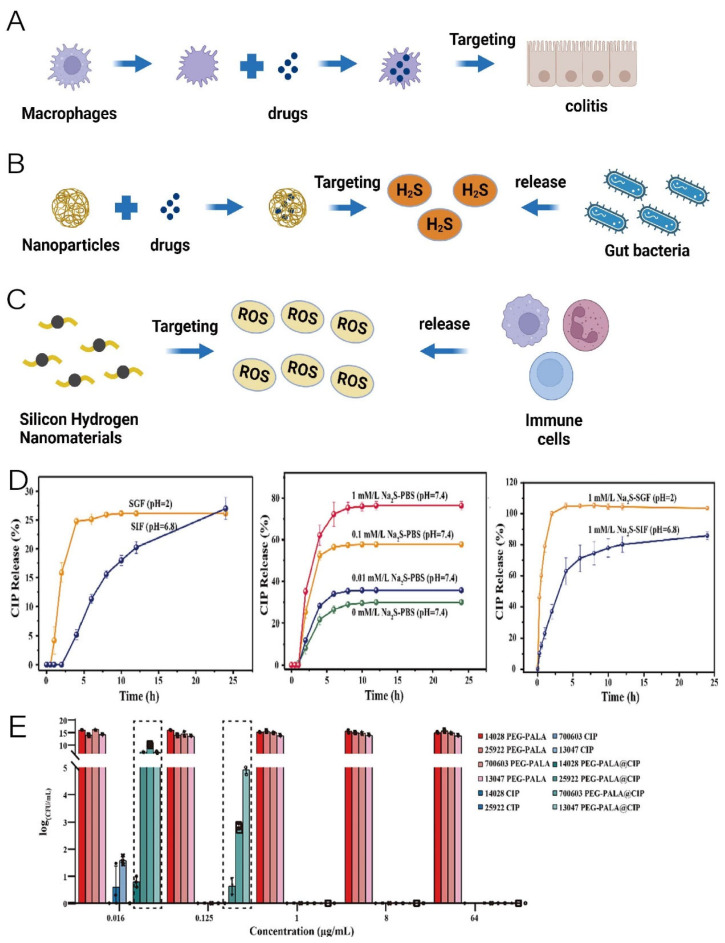
(**A**) Schematic representation of macrophage membrane-assisted drug targeting in colitis. (**B**) Schematic representation of nanomaterial targeting of H_2_S released by intestinal bacteria. (**C**) Schematic representation of silicon-based nanomaterial targeting of ROS released by immune cells. (created with bioRender.com) (**D**) Time-dependent release curves of ciprofloxacin under different conditions (CIP: ciprofloxacin, SGF: stimulated gastric fluid, SIF: stimulated intestinal fluid). (**E**) Bactericidal effect of encapsulated ciprofloxacin and free ciprofloxacin under different conditions (PEG-PALA: PEGylated poly (α lipoic acid), PEG-PALA@CIP: PEG-PALA nanoparticles encapsulated with ciprofloxacin). Adapted with permission from ref. [36]. Copyright 2022 Elsevier.

**Table 1 pharmaceutics-16-00921-t001:** Summary of strategies for nanomaterials as carriers to assist in drug modulation of intestinal microbiota.

Target	Material	Drugs	Therapeutic Effects	References
Sustained release	Nanoparticles based on Hohenbuehelia serotina polysaccharides and mucin	Polyphenols from Malus baccata (Linn.) Borkh	Improve and maintain intestinal health Optimizing the composition of gut microbiota.	[24]
	Self-assembled starch nanoparticles derived from quinoa, maize, and waxy maize starches	Chrysin and rutin (natural polyphenols)	Sustain the release of natural polyphenols Antioxidant activity	[26]
	Nanoparticles based on bovine serum albumin and Hohenbuehelia serotina polysaccharides	Polyphenols isolated from the shells of Juglans regia L.	Modulate the composition of gut microbiota Promote the production of short-chain fatty acids	[27]
	Nanostructured lipid carriers and nanostructured lipid carriers imbedded microcapsule	6-gingerol (polyphenols)	Antioxidant Anti-inflammatory Modulate the composition of gut microbiota Promote the intestinal barrier function	[28]
	A bioadhesive liquid coacervate based on bidentate hydrogen bonding-driven nanoparticle assembly	A water-soluble sodium phosphate salt of dexamethasone	Significantly mitigate the symptoms of inflammatory bowel disease Restore the diversity of gut microbiota Reduce systemic drug exposure Improve the therapeutic efficacy	[29]
Controlled release	Low molecular weight chitosan and unsaturated alginate resulting nanoparticles	Curcumin–cyclodextrin inclusion complex	Promote colonic epithelial barrier integrity Modulate the production of inflammatory cytokines Reshape the gut microbiota	[32]
	A novel nitroreductase labile peptidic hydrogel based on supramolecular self-assembly	Escherichia coli Nissle 1917 (probiotic)	Downregulate proinflammatory cytokines Repair the intestinal barrier Increase the diversity and abundance of indigenous probiotics	[34]
Targeted delivery	Encapsulation of nanocomplexes formed by the coupling of polydopamine nanoparticles with mCRAMP (mouse cathelicidin-related antimicrobial peptide) using macrophage membrane	mCRAMP (an antimicrobial peptide)	Reduce the secretion of pro-inflammatory cytokines Elevate the expression of anti-inflammatory cytokine Positive regulation of intestinal microbiota	[35]
	The PEGylated poly (α lipoic acid) (PEG-PALA) copolymer nanoparticles	Ciprofloxacin	Reduce the *Salmonella* colonization Prolong ciprofloxacin persistence in the intestine Keep the gut microbiota homeostasis	[36]
	Fe@Fe_3_O_4_ nanoparticles	Ginsenoside Rg3	Significantly inhibit hepatocellular carcinoma (HCC) development and eliminate HCC metastasis to the lung Correct tumor-dominant metabolomics Regulate the composition of the intestinal microbiota	[37]
	Rhamnolipid	Fullerene	Promote the colonization of intestinal probiotics Suppress the biofilm formation of pathogenic bacteria Improve the organismal immune system	[21]
	Highly cross-linked polyphosphazene nanodrug developed by copolymerisation of curcumin and acid-sensitive units (4-hydroxy-benzoic acid (4-hydroxy-benzylidene)-hydrazide) with hexachlorotripolyphosph-onitrile	Curcumin	Downregulate several critical inflammatory cytokines (TNF-α, IL-1β, and IL-8)	[38]

### 2.4. Protective Effect of Nanomaterials on Gut Microbiota

It is well known that the targeted use of antibiotics to inhibit the proliferation of harmful bacteria in the gut is an important means of maintaining intestinal health. However, the use of antibiotics is not conducive to maintaining the balance of gut microbiota and can increase the formation and selection of antibiotic-resistant bacteria [40]. The use of nanomaterials can prevent disruption of the gut microbiota by antibiotics (Figure 5A). Positively charged polymeric nanoparticles with a glucosylated surface carrying antibiotics can improve the utilization of antibiotics in the small intestine, prevent antibiotics from contacting the gut microbiota, and thus maintain the stability of the gut microbiota [41]. Microbial infiltration in tumors is related to the development and treatment of tumors. Self-assembly of an amphiphilic small molecule, metronidazole-fluorouracil into nanomaterials can target tumors, not only killing bacteria in tumors but also preventing disruption of intestinal microbiota by antibiotics [42]. Chunxiao Gao et al. collected feces from treated mice with colorectal cancer and conducted 16S rDNA sequencing. The results of Shannon, Ace, Sobs, Chao, and Simpson indices were analyzed as shown in Figure 5B. The community diversity and abundance of the MTI (metronidazole) group and the MTI and FDU (fluorouridine) co-administration group were significantly lower than that of the control group, whereas there was no significant change in the community diversity and abundance in the MTI–FDU nanomedicine group (Figure 5B). Principal coordinates analysis (PCoA) showed that the difference between the nanomedicine group and the control group was significantly smaller (Figure 5C). The results of community composition analyses showed a significant decrease in bacterial composition changes in the MTI–FDU nanomedicine group (Figure 5D) [42]. Drug-loaded electro-spun nanofibers can carry Streptomycin for delivery to the colon, reducing the risk of resistance and interference with the gut microbiota by antibiotics [22]. Lu Han et al. reported that the PEGylated poly (α lipoic acid) copolymer nanoparticles prolonged the retention time of drugs in the intestine through targeted delivery of antibiotics, thus significantly preventing disruption of intestinal microbiota stability by antibiotics [36].

## 3. Nanomaterials Regulate Intestinal Microbiota

In addition to serving as carriers, some nanomedicines can themselves intervene in disease processes by modulating the gut microbiota (Figure 6A). Yonghyun Lee et al. reported a nanosynthetic drug composed of hyaluronic acid and bilirubin, which enriches the diversity of gut microbiota and significantly increases the abundance of probiotic strains such as *Akkermansia muciniphila* and *Clostridium XIVα*, thereby alleviating inflammatory bowel disease (IBD) [43]. Yonghyun Lee et al. collected treated colitis mice fecal samples and analyzed them by 16s rRNA sequencing. The results of the analyses showed that HABN treatment significantly improved the diversity and composition of the intestinal microbiota of the mice (Figure 6B–D). Further analysis at the phylum/family level showed that HABN (hyaluronic acid-bilirubin nanomedicine) treatment significantly increased the relative abundance of probiotics such as *Akkermansia muciniphila*, *Clostridium XIVα*, and *Lactobacillus* (Figure 6E) [43]. Diwei Zheng et al. developed a biotic–abiotic hybrid nanosystem consisting of bacteriophages covalently linked to dextran nanoparticles encapsulated with irinotecan, which improved chemotherapy for colorectal cancer. This hybrid nanosystem not only inhibits the proliferation of *Enterococcus faecalis* in the gut but also promotes the growth of endogenous butyrate-producing bacteria [44]. Selenium (Se) nanoparticles can decrease the *Firmicutes/Bacteroidetes* ratio and increase the abundance of beneficial bacteria (e.g., *Akkermansia, Muribaculaceae, Bacteroides*, and *Parabacteroides*), thus protecting against acute toxicity induced by herbicide [45]. Hydrogen nanoparticles synthesized by encapsulating ammonia borane into hollow mesoporous silica nanoparticles can increase the abundance of *Akkermansia muciniphila* and reshape the gut microbiota, thereby improving metabolic disorders associated with fatty liver disease [46]. Rhamnolipid/fullerene-60 nanomaterials can promote the proliferation of probiotic bacteria such as *Lactobacillus, Bifidobacterium,* and *Enterococcus* in the gut while inhibiting the formation of biofilms by pathogenic bacteria such as *Escherichia coli* and *Shigella*, thereby alleviating ulcerative colitis [21]. Oral administration of silica-hydrogen nanoparticles can restore gut dysbiosis induced by dextran sulfate sodium (DSS), increasing the relative abundance of the beneficial bacteria *Muribaculaceae* and *Lachnospiraceae_NK4A136_group* while decreasing *Enterobacteriaceae* abundance [39]. Surface-deacetylated chitin nanofibers (SDACNF) can increase the abundance of beneficial bacteria from the genus *Blautia* while reducing the abundance of harmful bacteria from the genera *Morganella* and *Prevotella* in spontaneously hypertensive rats [47].

Furthermore, polyphenols, in addition to acting as drug carriers, can also serve as components of nanomaterials to help regulate the gut microbiota. Polyphenols, proteins, and nucleic acids have self-assembly capabilities. Bing Hu et al. assembled proteins into flexible biocolloids, and amyloid fibrils with high aspect ratios, which support the deposition of polyphenols and self-assemble with polyphenols into hybrid nanofibers. These hybrid nanofibers ultimately form functional macroscopic hydrogels. This hydrogel enhances the stability of polyphenols and, when orally administered, can remain in the colon, where the gut microbiota is most abundant, promoting intestinal barrier function, modulating the composition of gut microbiota, and alleviating metabolic disorders associated with colitis [48]. Resveratrol is a polyphenol. The resveratrol-selenium-peptide nanocomposite can regulate the dysbiosis of gut microbiota induced by Alzheimer’s disease, particularly oxidative stress and inflammation-related bacteria [49]. Artificial probiotics constructed by Jiaqi Xu et al. can increase the diversity of gut microbiota and regulate the microbial spectrum of the gut, thereby correcting dysbiosis in a colitis model [50]. Nanoparticles self-assembled from berberine and baicalin, traditional Chinese medicines, can reduce the abundance of *Bacteroidia, Deferribacteres, Verrucomicrobia, Candidatus_Saccharibacteria,* and *Cyanobacteria* in the gut of a diarrhea-predominant irritable bowel syndrome model [51].

Exosomes are natural nanovesicles secreted by cells, plants, and microorganisms, containing small molecules that contribute to human health maintenance. Exosomes derived from *Portulaca oleracea L* can promote the growth of *Lactobacillus reuteri* in the gut of dextran sulfate sodium (DSS)-induced colitis mice models [52]. Extracellular vesicles derived from *Clostridium butyricum* can significantly reduce the abundance of pathogenic bacteria such as *Escherichia coli* and *Shigella flexneri*, thereby alleviating ulcerative colitis in mice [53]. Pectins and modified pectins derived from marine microalgae and Spirulina maxima can increase the abundance of *Bacteroidetes* in the gut while reducing the abundance of *Firmicutes* [54].

## 4. The Dual-Edged Sword Effect of Nanomaterials on Gut Microbiota

Nanomaterials, due to their numerous advantages, have been increasingly utilized in various fields. However, with the growing exposure of nanomaterials in the living environment, scholars have become aware that nanomaterials are not flawless, particularly in their disruption of the human gut microbiota (Figure 7A). Exposure to graphene materials can significantly alter the human gut microbiota and disrupted gut microbiota, unfavorable for maintaining intestinal homeostasis, can further lead to adverse conditions [55]. Microplastics or nanoplastics are present in food and food packaging and are often ingested by humans through ingestion or inhalation. In mouse models, ingestion of microplastics or nanoplastics can lead to gut dysbiosis and immune imbalance. Although evidence of their toxicity in humans is currently lacking, we should remain vigilant about their potential risks to human health [56]. Studies have shown that nanoplastic ingestion can lead to gut microbiota and metabolic disorders, resulting in neurotoxicity via the gut–brain axis [57]. Furthermore, anxiety-like behaviors induced in mice by nanoplastic and microplastic exposure are also associated with gut microbiota dysbiosis [58]. To verify this finding, Xuebing Chen et al. exposed mice to PS-NPs (polystyrene nanoparticles) and PS-MPs (polystyrene microparticles) for 30 and 60 days and then performed open field tests (OFTs) and elevated plus maze tests (EPMs). The results showed that the PS-NPs and PS-MPs groups exhibited increased anxiety behavior compared to the control group (Figure 7B,C) [58].

Engineered nanomaterials, such as titanium dioxide (TiO_2_), silver (Ag), zinc oxide, and silica nanoparticles, are increasingly used in food and food packaging. Ingestion of nanomaterials entering the gut through accidental ingestion causes disruption of the gut microbiota, inhibition of probiotic proliferation, and alteration of the *Firmicutes/Bacteroidetes* ratio [59,60,61]. Robert Landsiedel et al. demonstrated that oral administration of silver nanoparticles significantly reduced the abundance of *Akkermansia* (a probiotic) in the gut and plasma levels of indole-3-acetic acid derived from gut microbes [62]. Moreover, when the gut microbiota composition is already compromised, such as by antibiotic use, it further exacerbates the intestinal damage caused by silver nanoparticles [63]. Exposure to zinc oxide nanoparticles reduces the abundance and diversity of intestinal microbiota and the ratio of probiotic to pathogenic bacteria [64]. Ingestion of food-grade titanium dioxide by young mice leads to dynamic changes in the gut microbiota (including the abundance of *Bacteroidetes, Lactobacillus, Bifidobacterium,* and *Prevotella*), which is one of the important reasons for titanium-dioxide-induced gut toxicity [65]. Only minor effects on the intestinal microbiota were observed with short-term (48 h) exposure to TiO_2_ nanoparticles; however, toxic effects on the intestinal microbiota were clearly observed with prolonged (28 days) exposure to TiO_2_ nanoparticles, such as an increase in the abundance of *Actinobacteria* and *Proteobacteria* as well as a decrease in the proportions of *Bacteroidetes* and *Firmicutes* [66]. NanoAl_2_O_3_ disrupts the gut microbiota in a colon simulator, reduces microbial metabolism, and increases pathways related to human gut microbiota and disease [67]. Dalel Askri et al. showed that oral administration of iron oxide nanoparticles did not affect the overall health of the animals, but showed a decrease in the relative weights of organs such as the brain, kidneys, lungs, and stomach, which may reflect the potential toxicity of iron oxide nanoparticles to humans [68].

Carbon nanotubes are novel nanomaterial that may induce immune toxicity. Inhalation of carbon nanotubes disrupts the composition of gut microbiota, increases the *Firmicutes/Bacteroidetes* ratio, and promotes inflammation. Exposure to both carbon nanotubes and cigarette smoke exhibits an additive effect on disrupting gut microbiota and promoting inflammation [69]. Furthermore, it has been shown that repeated exposure to multiwalled carbon nanotubes does not cause changes in the intestinal microbiota of mice [70]. On the contrary, multi-walled carbon nanotubes exacerbated the doxorubicin-induced changes in the intestinal microbiota [71]. High-dose injection of biomimetic mesoporous polydopamine nanoparticles disrupts gut microbiota, immune balance, and oxidative stress damage [72]. Poly(lactic-co-glycolic acid) (PLGA) nanoparticles used for scaffold implants can induce cecal acidification, reduce *Firmicutes* and *Bacteroidetes*, and disrupt microbial composition [73]. Ingestion of nanoparticles of polyethylene terephthalate (PET) also disturbs gut microbiota and intestinal metabolism [74]. Careful selection of nanomaterials and greater use of low-toxic or non-toxic nanomedicines can greatly advance the application of nanotechnology in regulating gut microbiota.

## 5. Nanomaterials Treat Diseases by Regulating the Intestinal Microbiota

### 5.1. Gastrointestinal Tract

#### 5.1.1. Immune Diseases

Inflammatory bowel disease (IBD) refers to a group of nonspecific chronic gastrointestinal inflammatory diseases of unknown etiology, including ulcerative colitis, Crohn’s disease, and indeterminate colitis. The incidence and prevalence of IBD are rapidly increasing worldwide, particularly in emerging industrialized regions such as East Asia [75]. Among these, East Asia has experienced the largest increase in disease burden, with the highest incidence of IBD occurring in the 30–34 age group and the peak prevalence in the 45–49 age group [76]. Additionally, in most countries worldwide, the incidence and prevalence of IBD in children continue to rise [77]. IBD is commonly associated with genetic susceptibility, impaired intestinal barrier function, dysbiosis of gut microbiota, and immune dysregulation (Figure 8) [78]. The combination of these factors contributes to the occurrence of IBD, although none of them individually are sufficient to cause colitis. IBD is a lifelong inflammatory disease, necessitating long-term medication to prevent disease progression [78]. The primary treatment for IBD is medication, with current conventional therapies including anti-tumor necrosis factor agents, 5-aminosalicylic acid (5-ASA), corticosteroids, and immunomodulators. Other medications include some novel small molecule drugs and biologics, such as selective Janus kinase inhibitors, PDE4 inhibitors, and IL-12/IL-23 inhibitors [79].

Although there is a lack of corresponding research results to demonstrate the causality between disrupted gut microbiota composition and IBD, there is indeed a correlation between the two [80]. Patients with IBD often exhibit changes in gut microbiota, including reduced microbial diversity, decreased stability of microbial composition, decreased abundance of *Firmicutes*, and proliferation of *Proteobacteria* [81]. Current research on gut microbiota mainly comes from the analysis of microbial changes in feces. Dari Shalon et al. developed an ingestible device that can sample from different parts of the intestine. They used this device to collect intestinal samples from healthy individuals for multi-omics analysis, demonstrating significant differences in bacteria, phages, host proteins, and metabolism between the intestine and feces [82]. However, studies have also shown that there are changes in microbial communities in the mucosa of the intestines with and without inflammation in the same IBD patients, as well as in mucosal biopsies of CD (Crohn’s disease) patients with and without damage [80]. In summary, there is a close relationship between IBD and dysbiosis of gut microbiota, and rebuilding the homeostasis of gut microbiota is a new strategy for alleviating IBD. New methods such as fecal microbiota transplantation, antibiotics, probiotics, and microbial metabolite inhibitors have been proposed. However, these treatments either have side effects or cannot achieve complete clinical relief when used alone [83].

One major challenge in the treatment of gastrointestinal inflammatory diseases is how to make drugs remain in the intestine for a long time to exert their therapeutic effects. The effective application of specific nanomaterials has overcome this challenge, and oral intervention with nanomaterials to modulate gut microbiota and alleviate IBD may have become a new treatment modality. Pengchao Zhao et al. reported a hydrogen-bond-driven nanoparticle-assembled bio-adhesive liquid coacervate, which exhibited significant structural stability and prolonged release of preloaded water-soluble small molecules for over two days. Compared to simple oral administration, orally administered drug-loaded nanoparticles significantly restored microbial diversity and alleviated symptoms of IBD [29]. Similarly, the use of nanomaterials has also greatly facilitated the use of polyphenols to intervene in IBD because nanomaterials significantly enhance the utilization of polyphenolic substances in the intestine. Wenni Tian et al. developed a 6-gingerol-loaded delivery system in the form of nanostructured lipid carriers (NLCs) or NLC-imbedded microcapsules (6G-MCs). 6G-NLCs alleviate colitis by modulating the *Firmicutes/Bacteroidetes* ratio, while 6G-MCs continuously release near the colonic lesion site and treat colitis by maintaining gut microbiota composition and diversity [28].

Furthermore, metabolic products of inflammatory cells such as ROS can damage intestinal cells and reduce their antioxidant capacity, which is also associated with the formation of IBD [21]. Yuxuan Xia et al. reported a therapeu**ti**c strategy for colitis using orally administered low-dose rhamnolipid (RL)/fullerene (C_60_) nanocomposite materials. The significant pH responsiveness endowed this material with excellent colon-targeting ability. The ROS scavenging ability of the C_60_ component effectively restored the anti-inflammatory capacity of colonic cells. Additionally, low-dose oral (RL)/C_60_ exhibited a remarkable ability to reconstruct the abnormal gut microbiota composition in colitic mice [21]. Similarly, Meiyu Bao et al. conjugated polydopamine nanoparticles with the antimicrobial peptide mCRAMP and encapsulated them with macrophage membranes to design a ROS-clearing and inflammation-targeting nanomedicine. This medicine not only regulated the gut microbiota of IBD patients and improved inflammatory responses but also demonstrated significant targeting effects in local inflammatory tissues due to macrophage membrane encapsulation [35]. Other studies have shown that clearing ROS can effectively improve intestinal inflammation in IBD. Importantly, ROS-targeting materials endowed nanocompounds with excellent lesion-targeting abilities, allowing drugs to accumulate at the lesion site [39,84,85,86]. Additionally, reactive nitrogen species produced in active mucosa can damage intestinal mucosa and barriers, accelerating the occurrence of IBD. Modified bismuth selenide polyvinylpyrrolidone (PVP) nanosheets, a multifunctional nanozyme based on 2D nanomaterials, can alleviate intestinal inflammation by clearing various reactive oxygen and nitrogen species [87].

By using probiotics, prebiotics can easily alter the structural composition of gut microbiota. Using nanomaterials to deliver probiotic products for disease treatment can not only enhance the activity and viability of probiotics reaching the target site but also promote targeted delivery of probiotics. Therefore, incorporating prebiotics or probiotics into nanomaterials has become a new treatment strategy for diseases [88,89,90]. Abdullah Glil Alkushi et al. combined dietary nanosupplements with *Bacillus* strains to intervene in colitis models, and their results showed that compared to free Bacillus spores, nanomaterial-loaded *Bacillus* significantly improved the therapeutic effect on colitis [90]. To further explore the feasibility of nanomaterial-combined probiotic strategies, Abdullah Glil Alkushi et al. integrated multispecies probiotics into nanomaterials and applied them to colitis model mice, ultimately demonstrating a better therapeutic effect on colitis than free multispecies probiotics [89]. Similarly, polydopamine nanoparticles can also encapsulate probiotics, which, when administered orally, can regulate gut microbiota and alleviate colitis [91]. Additionally, probiotic membranes are also promising nanomaterials. Jiaqi Xu et al. constructed artificial probiotics by using *Escherichia coli Nissle 1917* membrane (EM) as the surface and the biodegradable diselenide-bridged mesoporous silica nanoparticles bridged by disulfides (SeM) as the core. This nanomedicine can regulate the oxidative stress and inflammatory balance in the intestines of IBD mice, increase gut microbiota diversity, and modulate microbial spectrum [50].

Extracellular vesicles (EVs) are double-layered vesicular nanoparticles with a size ranging from 20 to 300 nm, encapsulating exosomes and micro vesicles, containing various small molecules, including proteins, lipids, and nucleic acids [53,92,93]. Human cells can secrete EVs, which can be extracted from various human body fluids, such as serum, urine, and cerebrospinal fluid. Additionally, the gut microbiota in humans and plants can also produce EVs [53,93,94]. EVs from various sources participate in the interaction and communication between intestinal epithelial cells, immune cells, and gut microbiota, thereby regulating intestinal inflammation and homeostasis (Figure 9A) [94]. EVs from different sources can affect the progression of IBD to some extent through different mechanisms. *Clostridium butyricum* has been shown to have a protective effect on intestinal inflammation. Research by Lingyan Ma et al. further demonstrated that the EVs secreted by these probiotics utilize the contained microRNA to inhibit inflammatory signaling pathways in the intestine. Additionally, these EVs can repair the dysbiosis caused by IBD, reduce the abundance of bacterial pathogens, and regulate tryptophan metabolism [53]. EVs secreted by other probiotics can also alleviate IBD. For example, EVs secreted by *Lactobacillus plantarum Q7* can alleviate IBD by modulating gut microbiota [95]. EVs derived from the symbiotic bacterium *F. prausnitzii* can alleviate IBD by regulating intestinal barrier function and immune function [96]. EVs derived from *Bacteroides thetaiotaomicron* can affect intestinal immune cells through different targets in healthy individuals and IBD patients [97]. Furthermore, EVs secreted by human mesenchymal stem cells can inhibit lipid peroxidation and iron death by targeting acyl-CoA synthetase long-chain family member 4 (ACSL4) and *miR-129-5p*, reducing intestinal inflammatory damage [98]. *Turmeric*-derived nanoparticles can inhibit intestinal inflammation, repair colonic damage, and possess colon-targeting ability to prevent and alleviate colitis [99]. An attenuation of DSS-induced inflammatory signaling in colitis was clearly observed after turmeric-derived nanoparticles (TDNP_S_2) intervention in mice, as shown by the results of chemiluminescent imaging (Figure 9B). In addition, MPO (myeloperoxidase) levels and expression of inflammatory factors, such as TNF-α, IL-6, and IL-1β, were reduced in colonic tissues, whereas the expression of oxidative stress-related protein HO-1 (heme oxygenase-1) was elevated. All these results showed the alleviating effect of TDNP_S_2 on colitis in mice (Figure 9C) [99]. Oral administration of edible mulberry bark-derived nanoparticles can activate the aryl hydrocarbon receptor/Constitutive Photomorphogenic Homolog Subunit 8 (AhR/COPS8) pathway mediated by plant heat shock protein heat shock protein family A (Hsp70) member 8 (HSPA8) to prevent DSS-induced colitis and promote the repair of intestinal microbiota composition [100]. Exosomes derived from *Portulaca oleracea L* can alleviate DSS-induced mouse models by reprogramming CD4+ T cells into CD4CD8 double-positive T cells, increasing levels of intestinal probiotics and indole derivatives [52]. In summary, EVs from different sources, including effector molecules, can affect the progression of IBD. Their protective ability, targeting ability, and low toxicity make them a natural nanocarrier, and EVs are more advantageous than other nanomaterials in terms of high yield and easy extraction [94,99]. More importantly, this natural nanocarrier has made oral nucleic acid therapy a reality. Yunyue Zhang et al. applied EVs carrying anti-TNF-α siRNA to an IBD rat model, significantly alleviating intestinal inflammation in the rat model [101]. In addition to carrying nucleic acids, EVs can also carry protein substances such as interleukin 10 and interleukin 27, which can be used to alleviate intestinal inflammation in IBD models [102,103]. Furthermore, Jiawei Guo et al. used exosomes inserted with Golgi Glycoprotein 1 to carry Wnt agonist 1. After injecting them into IBD mice, due to the targeting effect of Golgi Glycoprotein 1, this nanocomposite successfully accumulated in the bones and successfully alleviated bone loss in the IBD mouse model by upregulating the Wnt/β-catenin pathway in bone marrow mesenchymal stem cells [104].

In addition to the above, nanocomposites formed by aggregation of hyaluronic acid and bilirubin can increase the overall richness and diversity of gut microbiota in IBD mice and significantly promote the proliferation of *Akkermansia muciniphila* and *Clostridium XIVα* [43]. Yu Xu et al. reported pH- and redox-responsive Angelica sinensis polysaccharide nanoparticles, which significantly improved the stability of intestinal microbiota and the level of short-chain fatty acids in ulcerative colitis (UC) mice. The dual responsiveness endowed them with excellent colon-targeting ability, increasing their concentration at inflammatory sites [105]. Yaping Liu et al. developed an oral radioprotector, ergothioneine hyaluronate gel, which can prevent radiation-induced gastrointestinal syndrome by regulating oxidative stress, inflammation, and gut microbiota [106]. Novel nanoparticles self-assembled from denatured human serum albumin and selenite salts can alleviate cisplatin-induced intestinal mucositis by restoring anti-inflammatory bacteria and reducing pro-inflammatory bacteria [107].

#### 5.1.2. Cancer

The discovery of the relationship between microbiota and cancer can be traced back thousands of years, and recent studies have delved deeper into the connection between the two [108]. Numerous studies have indicated the close association between gut microbiota and the occurrence, development, diagnosis, and treatment of tumors [109,110,111]. For instance, *Lactobacillus gallinarum* and *Akkermansia muciniphila* can prevent the occurrence of intestinal tumors, while *Lactobacillus rhamnosus* can enhance the efficacy of anti-tumor therapy [111,112,113]. Additionally, some metabolites related to gut microbiota contribute to the detection of colorectal cancer [110]. Therefore, modulating the gut microbiota has emerged as a novel strategy to improve cancer treatment outcomes. For example, Pien Tze Huang can prevent colorectal cancer by regulating the gut microbiota and metabolites [114]. Ginseng polysaccharides can enhance the efficacy of anti-tumor therapy by modulating the gut microbiota and tryptophan metabolism [115]. At the same time, nanomaterials serve as excellent media for modulating the gut microbiota, and their good intervention ability in the gut microbiota can significantly improve cancer treatment outcomes (Figure 10A) [116]. Utilizing a bio-inorganic hybrid system formed by bacteriophage and dextran nanoparticles to carry irinotecan can enhance the efficacy of first-line chemotherapy for colorectal cancer by modulating the proliferation of harmful bacteria and probiotics in the intestine, thus reducing the side effects of chemotherapy [44]. Tianqun Lang et al. constructed nanoparticles using a prebiotic xylan–stearic acid conjugate, which not only increases the intra-tumoral drug concentration by encapsulating capecitabine but also enhances chemotherapy efficacy by promoting probiotic proliferation and short-chain fatty acid production [117]. Qunlang Tian et al. performed in vivo bioluminescence imaging using an in vivo imaging system (IVIS) to demonstrate tumor formation in treated mice (Figure 10B). The results of variation in relative tumor volumes calculated according to the amount of photons showed that applying SCXN (a Cap-loaded nanoparticle using the prebiotic xylan–stearic acid conjugate) to colorectal cancer mice had a higher rate of tumor suppression compared to other groups (Figure 10C) [117]. Zhigang Ren et al. constructed a nanodrug using Fe@Fe_3_O_4_ coupled with ginsenoside Rg3, which increased the abundance of *Bacteroidetes* and *Verrucomicrobia*, reduced the abundance of *Firmicutes*, and significantly prolonged the survival time of mice with hepatocellular carcinoma [37]. Moreover, electrostatic assembly of silver nanoparticles (AgNP) onto the surface capsid protein of M13 bacteriophage(M13@Ag), which specifically binds to *Fusobacterium nucleatum* (Fn), can achieve specific clearance of Fn and reshape the tumor immune microenvironment. This nanocomplex, when used in combination with immune checkpoint inhibitors (α-PD1) or chemotherapy drugs, can significantly improve the overall survival of mice with primary colorectal cancer [118].

#### 5.1.3. Other Disease

Modulation of intestinal microbiota by orally administered nanomaterials can also be used as a therapeutic strategy for other gastrointestinal disorders. Nanocrystalline cellulose can promote the metabolism of bile acids, fatty acids, and amino acids by improving the composition of the gut microbiota, thereby alleviating constipation [119]. Exosome-like nanoparticles derived from broccoli can alleviate loperamide-induced constipation by restoring gut microbiota dysbiosis and regulating short-chain fatty acid and tryptophan metabolism [120]. Nanoparticles self-assembled from berberine and baicalin, traditional Chinese medicine components, can alleviate diarrhea-predominant irritable bowel syndrome in mice by regulating the composition of the gut microbiota [51]. Nanoparticles formed by gold nanostars and acid-sensitive cis-aconitic anhydride conjugates modified with anti-*Helicobacter pylori* polyclonal antibodies can be used for the diagnosis and treatment of *H. pylori* infection. Treatment doses of oral gold nanostars@H. pylori-antibodies nanoprobes (GNS@Ab) can target and eradicate *Helicobacter pylori* without disrupting the balance of intestinal microbiota [121]. Nanomaterial fullerol can inhibit bacterial translocation and neutrophil infiltration in mice undergoing ischemia-reperfusion, reducing intestinal damage caused by ischemia-reperfusion and delaying or reducing mortality [122]. In a weaned pig model, dietary supplementation with different concentrations of coated nano-zinc oxide (Cnz) can modulate the core composition of fecal bacteria and improve intestinal mucosal barrier function [123]. Low doses of Chitosan nanoparticles–microcin J25 (CNM) significantly increase the weight of mice infected with *Escherichia coli O157:H7* and have beneficial effects on lifespan or clinical symptoms, while significantly improving intestinal health [124]. Green carbon dots derived from Atractylodes macrocephala can effectively alleviate alcohol-induced gastric ulcers by regulating immunity, oxidative stress, gastric enzyme activity, and intestinal microbiota [125].

### 5.2. Extra-Intestinal Gastrointestinal Tract

#### 5.2.1. Immune-Related Diseases

The interaction between the digestive tract and the intestinal microbiota is essential for maintaining normal function and homeostasis of the gastrointestinal tract. In addition, intestinal bacteria and secretory products can pass through the gastrointestinal tract into the bloodstream to affect various organs of the body. Inflammation of extra-intestinal organs is often accompanied by abnormalities in the intestinal microbiota, so the regulation of the microbiota can to a certain extent alleviate inflammation in other parts of the body outside the gastrointestinal tract. Yuan Li et al. reported that ultrasmall cortex moutan nanoclusters can effectively treat infectious pneumonia and colitis [126]. Polyphenol-containing nanoparticles of curcumin polyphosphonate can treat acute lung injury and improve functional recovery after spinal cord injury [38,127]. Surface-deacetylated chitin nanofibers (SDACNF) can exert hepatoprotective and antioxidative effects by adsorbing lipid substances and regulating gut microbiota [47]. Nel-like Molecule Type 1 Combined with Gold Nanoparticles can treat periodontitis by modulating macrophage polarization and osteoclast activity and improving the periodontal microbiota. However, this medication disrupts the normal composition of the intestinal microbiota [128]. Dietary supplementation of chitosan nanoparticles in the diet can regulate the growth performance and immune status of weaned pigs as well as modulate the ratio of beneficial bacteria to harmful bacteria in the gut [129]. Licong Yang et al. utilized dihydromyricetin-encapsulated selenium nanoparticles to construct dihydromyricetin@ selenium nanoparticles (DMY@SeNPs), and further modified them stepwise with chitosan (CS/DMY@SeNPs) and blood-brain barrier (BBB)-targeting peptide Tg (TGNYKALHPHNG) to obtain Tg-CS/DMY@SeNPs. This nanomaterial can ameliorate neuroinflammation via the gut microbiota-NLRP3(nucleotide-binding oligomerization domain leucine-rich repeat and pyrin domain-containing protein 3) inflammasome-brain axis [130]. Sahar Y. Al-Okbi et al. found that the administration of basil essential oil nanoemulsion (BNO) and its parent basil essential oil (BO) in rats can improve liver lipid and histopathological changes in non-alcoholic fatty liver disease [131]. Obesity can induce chronic inflammation (including neuroinflammation) via the brain–gut axis. Kumaran Sundaram et al. discovered that garlic exosome-like nanoparticles can suppress systemic and cerebral inflammatory activity, and reverse obesity induced by a high-fat diet in mice [132].

#### 5.2.2. Metabolic-Related Diseases

Past studies have shown that the stability of the gut microbiota is likely to be closely related to the metabolic health of the body and that abnormalities in the gut microbiota are likely to be involved in the development of metabolic diseases such as obesity and alcoholic fatty liver disease. Currently, nanomaterials that intervene in the gut microbiota to improve metabolic health are a novel response to metabolic disorders (Figure 11). Amyloid β (Aβ) is a key protein in the pathogenesis of Alzheimer’s disease (AD), and the abnormal accumulation of Aβ in brain tissue is one of the important contributors to the development of AD. Resveratrol-selenium-peptide nanocomposites can alleviate Alzheimer’s disease in mice by inhibiting β-amyloid aggregation in the hippocampus, exerting antioxidative effects, maintaining neuronal homeostasis, and regulating gut microbiota [49]. Chiral Au nanoparticles can modulate the gut microbiota of Alzheimer’s disease model mice, increase indole-3-acetic acid in the intestines, and thus alleviate cognitive impairment in mice [133]. Mesoporous silica nanoparticle-encapsulated *Bifidobacterium* not only can suppress intestinal inflammation and reduce brain Aβ burden but also can enhance olfactory sensitivity in mice [134]. The pathology of Alzheimer’s disease (AD) is highly associated with obesity-induced insulin resistance. Licong Yang et al. utilized chitosan (CS) and sodium tripolyphosphate (TPP) to carry low water solubility Res to construct Res-loaded CS/TPP nanoparticles. After oral administration, these nanoparticles can alleviate lipid-deposition-induced insulin resistance, reduce the level of Aβ aggregation, and modulate the gut microbiota to improve cognitive impairment in obesity-related Alzheimer’s disease mice [135]. Gold nanospheres can mitigate estrogen deficiency-induced bone loss by regulating gut microbiota homeostasis and reducing trimethylamine-N-oxide (TMAO) metabolism [136]. TMAO is formed by the hepatic conversion of trimethylamine (TMA) derived from nutrient metabolism by intestinal microbiota, and its elevated concentration may lead to serious cardiovascular events [137]. Dietary supplementation with selenium nanoparticles can alleviate acute toxicity induced by diquat via modulating gut microbiota and its metabolism [45]. Zhaokui Jin et al. synthesized hydrogen nanoparticles by encapsulating ammonia borane into hollow mesoporous silica nanoparticles, which continuously release hydrogen gas in the intestine, reshaping gut microbiota and alleviating metabolic dysfunction-associated fatty liver disease [46]. Ginger nanoparticles can prevent insulin resistance in high-fat diet (HFD) mice by restoring the stability of intestinal epithelial *Foxa2*-mediated signaling [138].

## 6. Discussion and Outlook

As described in this paper, an increasing number of scholars recognize the importance of the gut microbiota in human health. Changes in the gut microbiota not only affect intestinal function but also impact various organs throughout the body through different pathways, such as the gut–brain axis, gut–bone axis, and gut–liver axis [133,136,139]. Treating diseases by influencing the gut microbiota has gradually become a novel therapeutic strategy. Conventional interventions targeting the gut microbiota mainly involve drugs, probiotics and prebiotics, and fecal microbiota transplantation, but these measures often have drawbacks. The use of drugs and probiotics/prebiotics is often limited by the intestinal structure and environment, while fecal microbiota transplantation may lead to many side effects, such as mechanical damage to the intestines, and may not be well accepted by patients. The use of nanotechnology can effectively overcome the shortcomings of microbiota-regulating drugs. Nanomaterials, as carriers, not only protect microbiota-regulating drugs from enzymatic and gastrointestinal degradation but also achieve controlled release and targeted delivery of drugs through responsiveness to pH, reactive oxygen species (ROS), or intestinal enzymes. Additionally, some substances themselves can serve as constituents of nanomaterials, and the resulting nanodrugs often exhibit superior effects compared to their individual components [48]. The combination of nanomaterials with microbiota-regulating drugs can greatly advance the development of gut microbiota-related therapies, such as those for inflammatory bowel disease (IBD), tumors, Alzheimer’s disease, etc.

Certainly, there are certain limitations to using nanomaterials to regulate gut microbiota. The biosafety of most existing nanomaterials for in vivo applications remains unclear, which poses significant challenges for the clinical translation of these materials. Meanwhile, we have also noticed that due to the continuous invention of numerous new materials, there is still a lack of research methods and evaluation criteria for the in vivo metabolic behavior of these new materials. When the human body is in a diseased state, it is often the result of various factors, and changes in gut microbiota are typically complex, involving disruptions in microbiota composition, metabolism, and other aspects. However, current research on using nanotechnology to regulate gut microbiota and alleviate diseases often only addresses a partial disruption to achieve the desired therapeutic effect. For example, using nanocarriers to deliver probiotics or antibiotics, or carrying other microbiota-modulating molecules. While such therapeutic strategies can indeed alleviate diseases, combining nanotechnology with multiple treatment methods to comprehensively regulate gut microbiota may achieve better therapeutic outcomes. Therefore, utilizing nanomaterials as a hub and incorporating various microbiota-regulating components as elements to construct nanocomplexes capable of comprehensively modulating gut microbiota for disease treatment may become one of the future research directions. However, compared to the former approach, the latter imposes higher demands on the performance of nanomaterials.

## Figures and Tables

**Figure 1 pharmaceutics-16-00921-f001:**
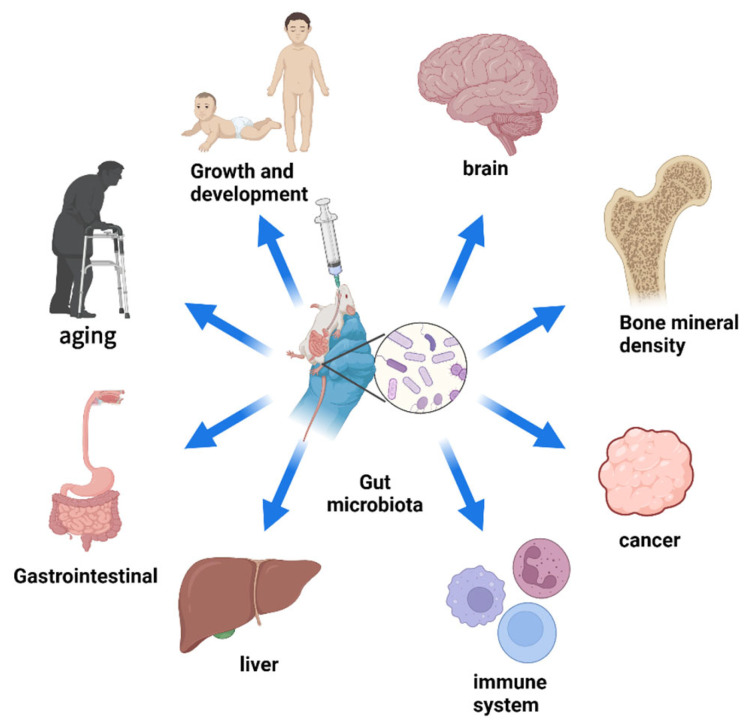
Schematic diagram of the relationship between gut microbiota and human physiology and diseases. (Created with bioRender.com).

**Figure 2 pharmaceutics-16-00921-f002:**
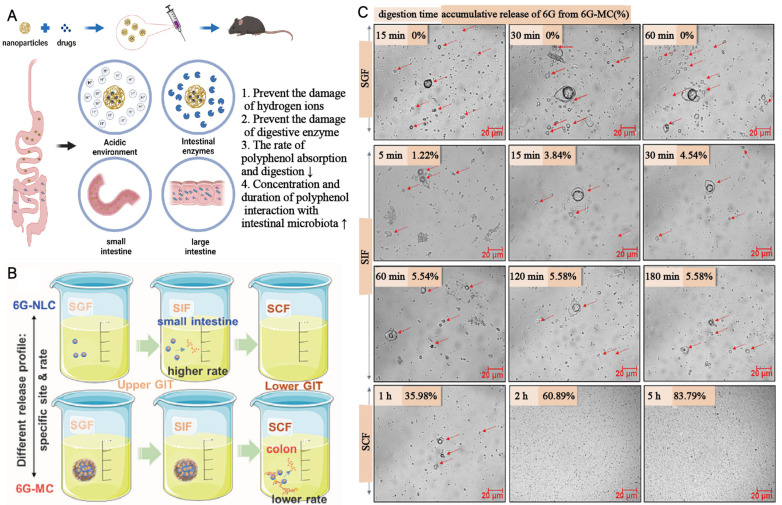
(**A**) Diagram illustrating the protection of drugs by nanomaterials (created with bioRender.com) (**B**) Schematic representation of the release of 6G from 6G-NLC and 6G-MC in different simulated digestive fluids (6G: 6-gingerol, NLC: nanostructured lipid carriers, MC: NLC-imbedded microcapsule, SGF: stimulated gastric fluid, SIF: stimulated intestinal fluid, SCF: stimulated colon fluid, GIT: gastrointestinal tract). The arrows in the figure point to 6G-MC in different states. This figure demonstrates the structural evolution of 6G-MC in different environments (**C**) Observation of the release of 6G from 6G-MC in different simulated digestive fluids under an optical microscope, accompanied by the cumulative release values of 6G at different time points. Adapted with permission from ref. [28]. Copyrights 2022 Elsevier.

**Figure 5 pharmaceutics-16-00921-f005:**
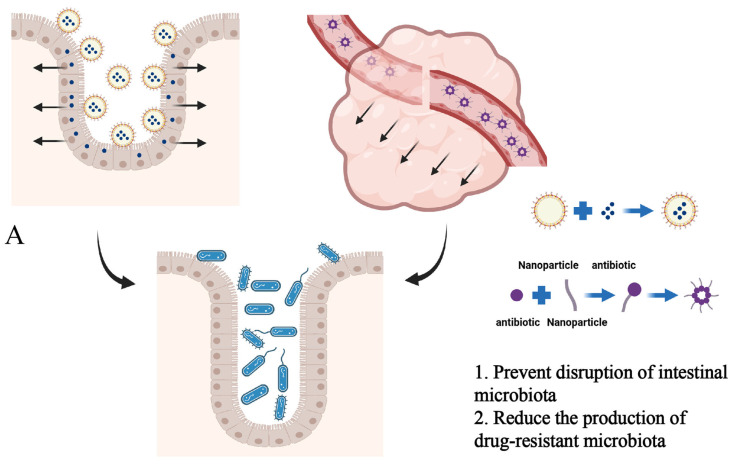
(**A**) Schematic representation of nanomaterials maintaining the stability of intestinal microbiota. (Created with bioRender.com) (**B**) Analysis of alpha diversity coefficients, Shannon index, Sobs index, Ace index, Chao index, and Simpson index of intestinal microbiota through 16S rDNA sequencing (Fn: *Fusobacterium nucleatum*, MTI: metronidazole, FDU: fluorouridine). (**C**) Principal coordinates analysis (PCoA) for beta diversity analysis. (**D**) Stacked bar plots showing the relative abundance of bacterial communities at the phylum and genus levels in samples from mice subjected to different treatments. Adapted with permission from ref. [42]. Copyright 2023 American Chemical Society.

**Figure 6 pharmaceutics-16-00921-f006:**
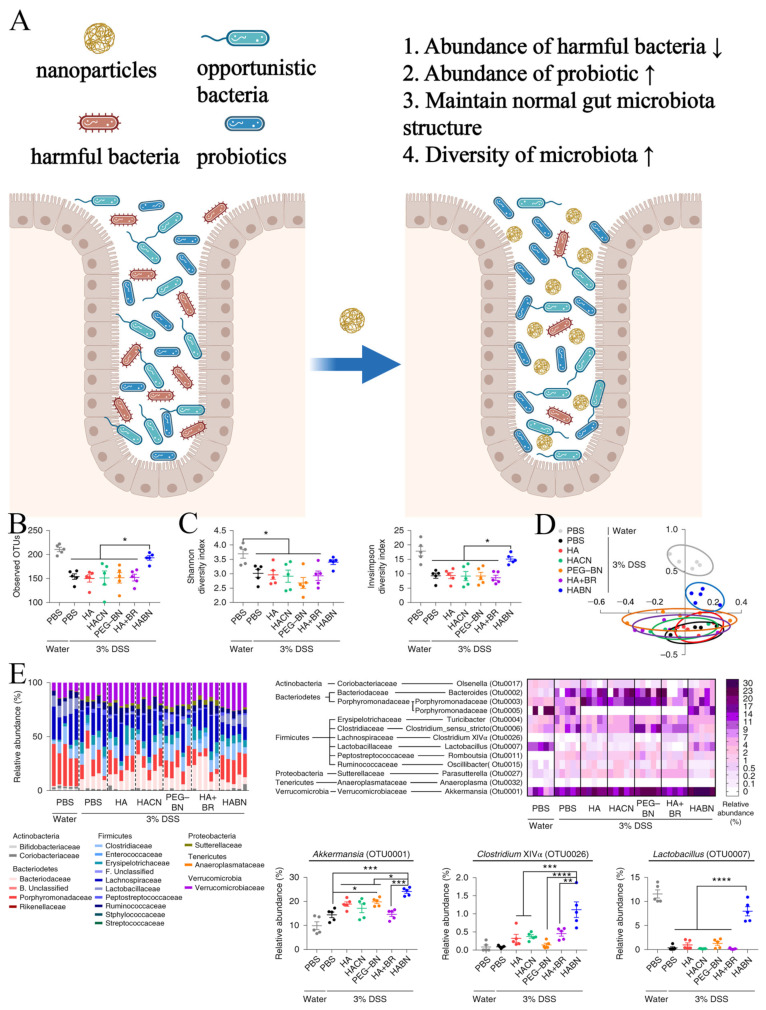
(**A**) Schematic representation of nanomaterials modulating intestinal microbiota (created with bioRender.com). (**B**–**E**) 16S rRNA sequencing analysis of mouse gut microbiota (HA: Hyaluronic acid, HACN: HA-cholesterol conjugates, PEG-BN: PEGylated bilirubin nanoparticles, HA + BR: HA and bilirubin, HABN: hyaluronic acid-bilirubin nanomedicine). (**B**) Abundance of operational taxonomic units (OTUs) in microbial communities. (**C**) Estimation of alpha diversity. (**D**) Beta diversity of gut microbiota. (**E**) Relative abundance of gut microbiota. Adapted from ref. [43] Copyright 2019 Nature Publishing Group. * *p* < 0.05, ** *p* < 0.01, *** *p* < 0.001, **** *p* < 0.0001, analyzed by one-way ANOVA with Tukey’s HSD multiple comparison post hoc test.

**Figure 7 pharmaceutics-16-00921-f007:**
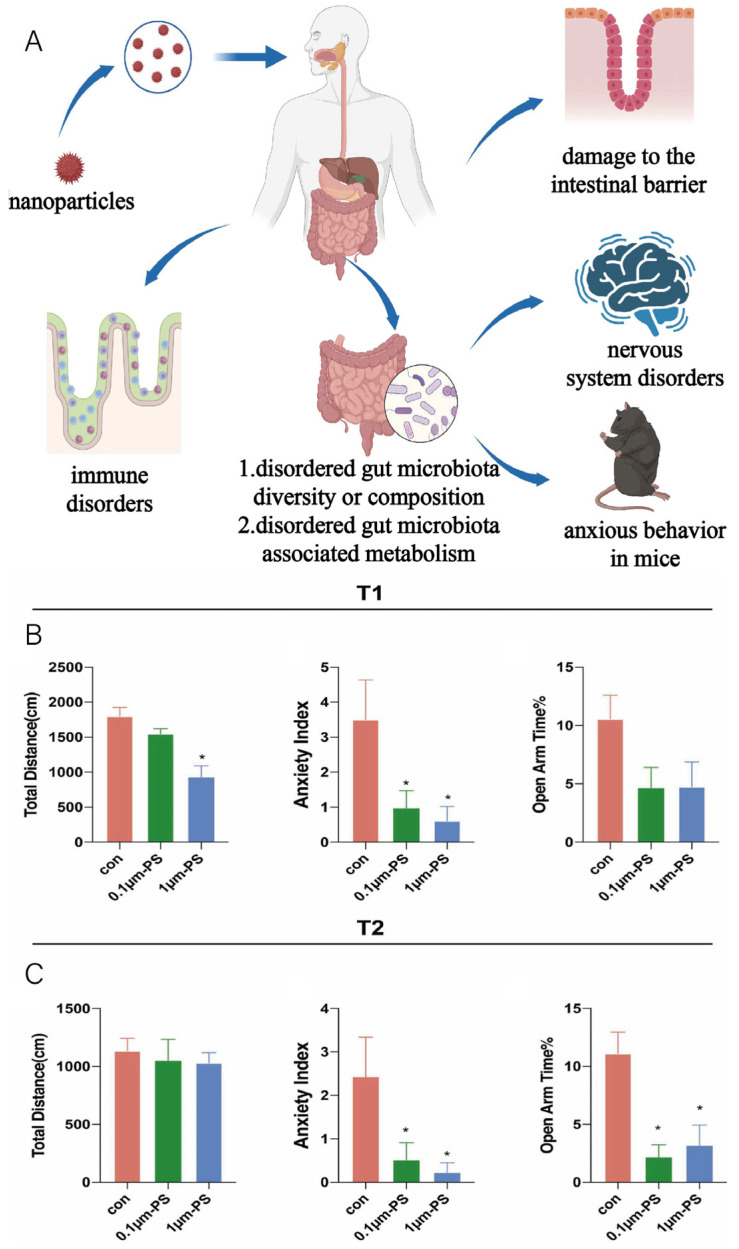
(**A**) Schematic representation of nanomaterials disrupting intestinal microbiota leading to disease (created with bioRender.com). (**B**,**C**) PS-MPs and PS-NPs induced anxiety in the adult mice (PS-MPs: polystyrene-microplastics, PS-NPs: polystyrene-nanoplastics). (**B**) Open field test (OFT) and elevated plus maze test (EPM) results for 30 days. (**C**) OFT and EPM results for 60 days. Adapted from ref. [58]. Copyright 2023 Academic Press Inc. Elsevier Science. * indicates a significant difference among the three groups (* *p* < 0.05).

**Figure 8 pharmaceutics-16-00921-f008:**
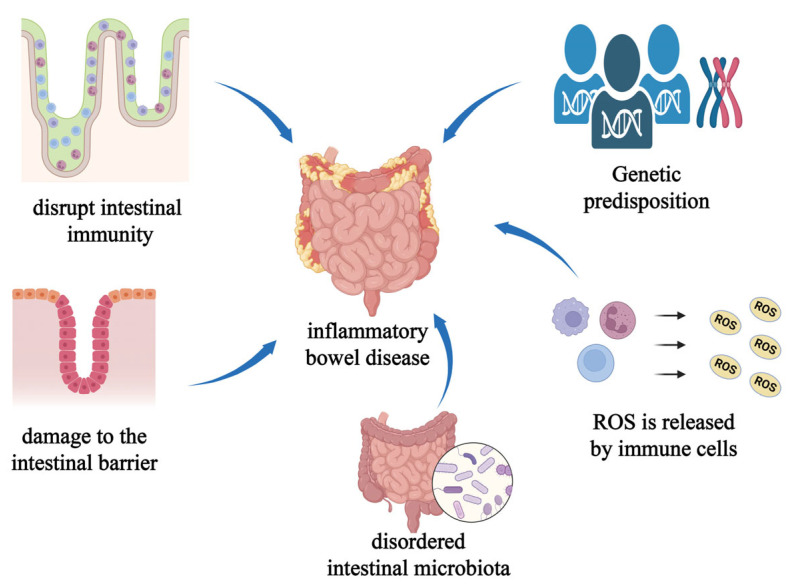
Schematic representation of factors associated with inflammatory bowel disease. (created with bioRender.com).

**Figure 9 pharmaceutics-16-00921-f009:**
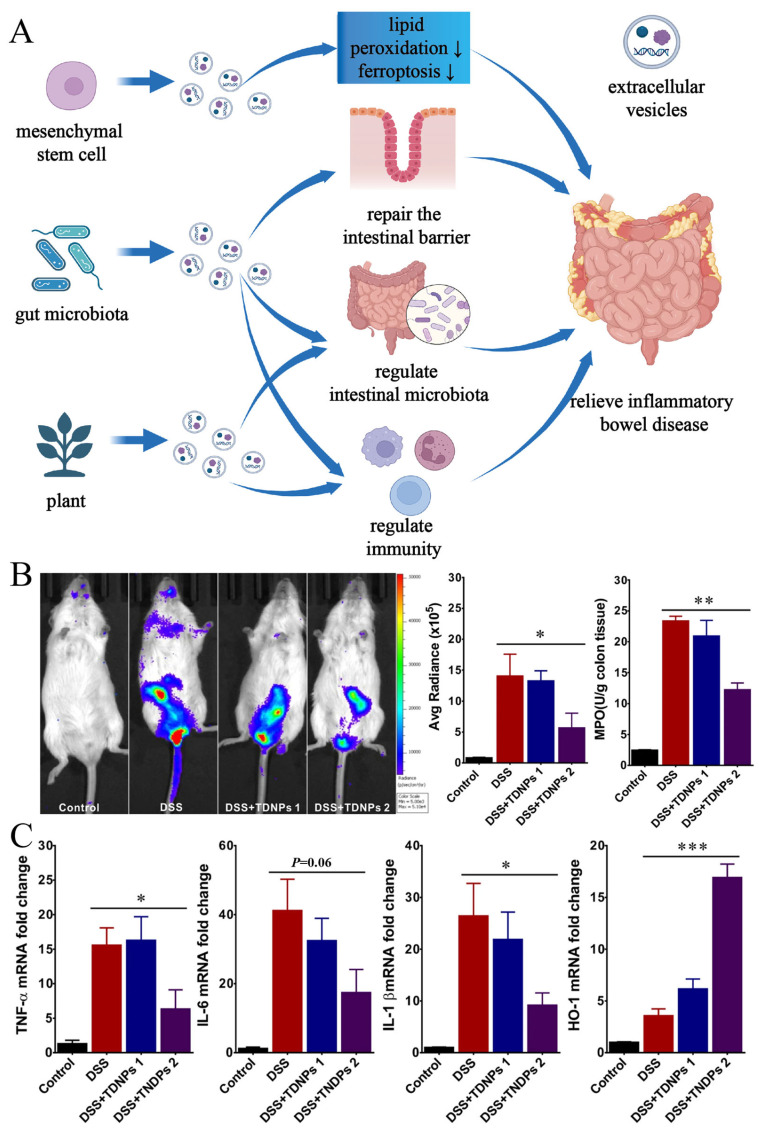
(**A**) Schematic representation of intervention of inflammatory bowel disease by extracellular vesicles from different sources. (created with bioRender.com) (**B**) Colon inflammation was monitored by XenoLight RediJect inflammation probe via chemiluminescence imaging (DSS: dextran sulfate sodium, TDNP: turmeric-derived nanoparticles, band 1 from the sucrose gradient interfaces of 8/30% was named TNDPs 1, and band 2 from the sucrose gradient interfaces of 30/45% was named TNDPs 2). (**C**) Real-time PCR to quantify miRNA. Adapted from ref. [99] Copyright 2022 BioMed Central Ltd. * *p* < 0.05, ** *p* < 0.01, *** *p* < 0.001 and ns represent no significant difference.

**Figure 10 pharmaceutics-16-00921-f010:**
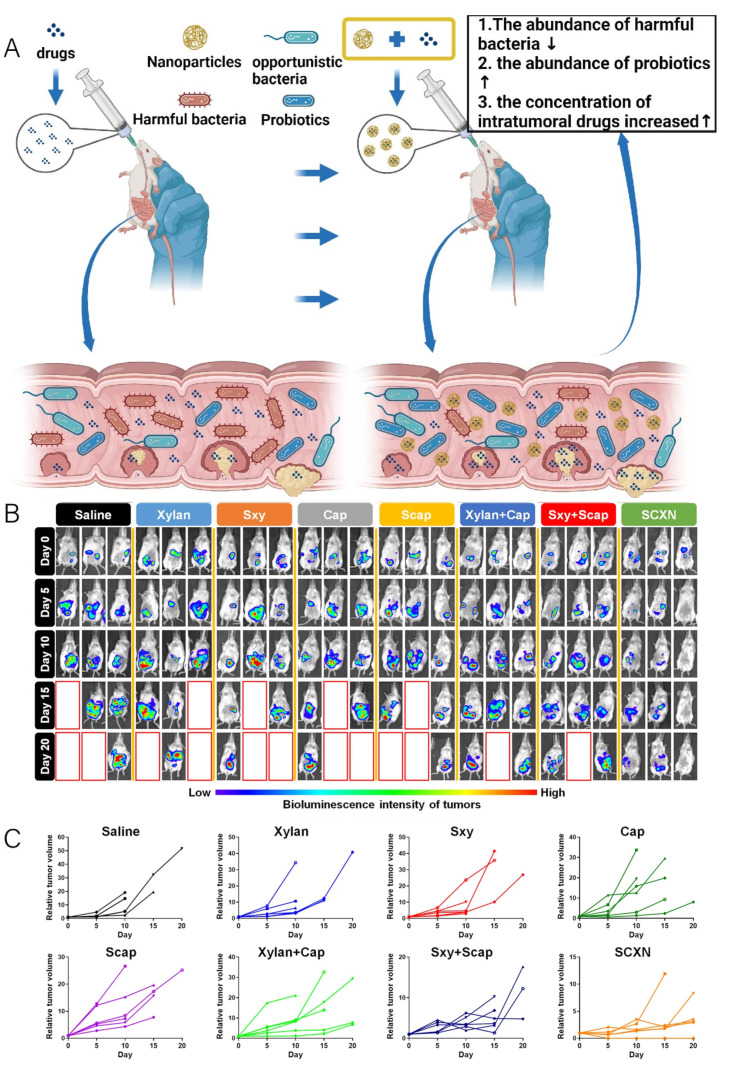
(**A**) Schematic representation of nanomaterial-assisted drug therapy for tumors (created with bioRender.com). (**B**) In vivo bioluminescence imaging using an in vivo imaging system (IVIS) (Cap: Capecitabine, Sxy: amphiphilic derivatives of stearic acid (Sa) and xylan, Scap: amphiphilic derivatives of stearic acid (Sa) and Cap, SCXN: a Cap-loaded nanoparticle using the prebiotic xylan-stearic acid conjugate). (**C**) Change in relative tumor volume calculated based on photon flux during the treatment period. Adapted from ref. [117]. Copyright 2023 Nature Publishing Group.

**Figure 11 pharmaceutics-16-00921-f011:**
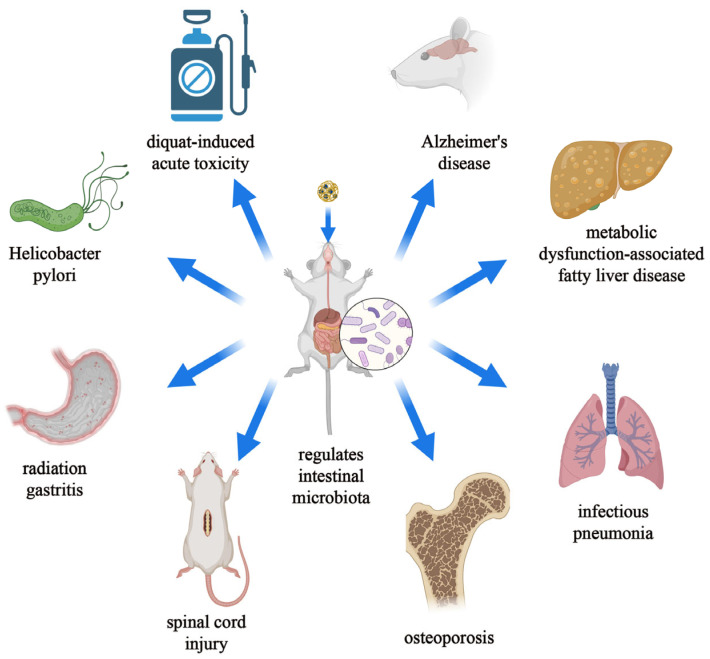
Schematic representation of nanomaterial-mediated alleviation of diseases by modulating gut microbiota. (created with bioRender.com).

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
