# Peer review of "Recent Progress of Oral Functional Nanomaterials for Intestinal Microbiota Regulation"

_pharmaceutics, 2024, doi:10.3390/pharmaceutics16070921_

Round 1

Reviewer 1 Report (Previous Reviewer 2)

Comments and Suggestions for Authors

No further criticism.

Author Response

Reviewer 2 Report (Previous Reviewer 3)

Comments and Suggestions for Authors

The authors significantly modified the work according to the reviewer's suggestions.

Currently, the main attention concerns the definition of gut/intestinal flora. This term is outdated. Flora refers to plants. Because we mean bacteria, viruses and protozoa, the term microbiota is used today.

Author Response

This manuscript is a resubmission of an earlier submission. The following is a list of the peer review reports and author responses from that submission.

Round 1

Reviewer 1 Report

Comments and Suggestions for Authors

Dear authors,

I revised the manuscript with ID pharmaceutics-2967835, which I found suitable to be published in the Pharmaceutics journal. The review provides a comprehensive overview of the importance of gut microbiota in human health and the advancements in using nanomaterials for microbiota regulation. It covers various aspects such as the role of gut microbiota, challenges in microbiota regulation, and the potential of nanomaterials in this field. The work is well-organized with a clear structure, including an abstract that summarizes the key points, introduction, methods, results, discussion, and conclusions. The review includes an in-depth analysis of specific studies and examples of nanomaterial applications for gut microbiota regulation, providing valuable insights into the potential benefits of using nanotechnology in this context.

There are also limitations to this work. The review briefly mentions limitations to using nanomaterials for microbiota regulation but does not delve deeply into potential challenges or drawbacks associated with this approach. A more thorough discussion of the limitations and potential risks of using nanomaterials in gut microbiota regulation will provide a more balanced perspective. The review focuses primarily on the benefits and advancements in using nanomaterials for microbiota regulation but it may benefit from a more comparative analysis of different approaches or technologies in this field. It could be strengthened by including a section on future directions or emerging trends in this area.

Reviewer 2 Report

Comments and Suggestions for Authors

This paper reviews data on the effects of different nanomaterials on the intestinal microbiota regulation.  This topic is obviously very important, both on the scientific and medical viewpoints, which deserves the interest for such a publication.

Nevertheless, I have severe criticisms, which according to me, impairs its publication in its present form.

1.     The scope is too broad and not specific enough.  For example, mixing of different therapeutical applications or biological concepts, sometimes in the same paragraph is puzzling. 

2.     To facilitate the reading and improve interest for the readers, all the figures with experimental data should be better discussed.  All abbreviations should be defined in the legend of the figure.  The authors should give minimal details on the structure and composition of the nanoparticles that are presented otherwise the interest for the readers will remain low.

3.     100 of the139 references - i.e. 72% – that are cited in this review have a Chinese scientist as first author.  The proportion surprises me because I doubt that almost ¾ of the scientific literature on this topic is published by laboratories from China, although I clearly consider that the contribution of these scientists is of great interest.  To become acceptable, the paper should be based on a worldwide review of the literature.

Reviewer 3 Report

Comments and Suggestions for Authors

The work presents attempts to use nanotechnology to modulate the microbiota.

The work is detailed, and the figures are an additional advantage.

However, there are a few minor issues that require attention:

- No reference to Figure 1 in the text

- I suggest replacing "senesence" in Figure 1 for "aging"

- All references in figures are either uppercase or lowercase

- I suggest "Gut microbiota diversity or composition" instead of "structure"

- Neurological or nervous system disorders instead of nervous disorders

- What do the Authors mean by "disorders of microbiota metabolism"?

- Have there been any attempts to reduce the concentration of TMAO, one of the best described microbiota metabolites?

- A good option would be a table summarizing nanotechnology strategies to improve the effects of microbiota-modulating compounds along with examples of application

- There are no limitations to the work - the lack of a systematic review of the literature and the subjectivity of the choice of the presented methods and their applications

Round 2

Reviewer 1 Report

Comments and Suggestions for Authors

the manuscript has been amended accordingly.

Reviewer 2 Report

Comments and Suggestions for Authors

The manuscript has been significantly improvred and more "international" references have been included.

Nevertheless, most experimental figures still need more explanations to become fully understandable… or be deleted.